# A Micro Electrochemical Sensor for Multi-Analyte Detection Based on Oxygenated Graphene Modified Screen-Printed Electrode

**DOI:** 10.3390/nano12040711

**Published:** 2022-02-21

**Authors:** Baiqing Yuan, Liju Gan, Gang Li, Chunying Xu, Gang Liu

**Affiliations:** School of Chemistry and Materials Science, Ludong University, Yantai 264025, China; gan17737233930@163.com (L.G.); 1258331922@126.com (G.L.); chunyingxu@126.com (C.X.)

**Keywords:** micro electrochemical sensor, multi-analyte detection, graphene oxide (GO), biofouling, oxo functionalities

## Abstract

Electrode interfaces with both antibiofouling properties and electrocatalytic activity can promote the practical application of nonenzymatic electrochemical sensors in biological fluids. Compared with graphene, graphene oxide (GO) possesses unique properties such as superior solubility (hydrophilicity) in water, negative charge, and abundant oxygenated groups (oxo functionalities) in the plane and edge sites, which play an essential role in electrocatalysis and functionalization. In this work, a micro electrochemical sensor consisting of GO-modified screen-printed electrode and PDMS micro-cell was designed to achieve multi-analyte detection with excellent selectivity and anti-biofouling properties by electrochemically tuning the oxygen-containing functional species, hydrophilicity/hydrophobicity, and electrical conductivity. In particular, the presented electrodes demonstrated the potential in the analysis of biological samples in which electrodes often suffer from serious biofouling. The interaction of proteins with electrodes as well as uric acid was investigated and discussed.

## 1. Introduction

Electrochemical sensing represents a potential tool to perform clinical applications due to its low cost, simplicity and high selectivity and sensitivity. However, when the electrodes are exposed to biological fluids such as plasma or blood, biofouling often occurs due to the non-specific absorption of biological macromolecules, especially proteins, which hinders the electron transfer and thus results in a rapid loss of sensitivity [1,2]. This issue could be solved for the glucose meter combination with a semi-permeable membrane, in which small-molecule glucose could diffuse through the semi-permeable membrane and reach the electrode, but not for macromolecule proteins [3]. In addition to the enzyme and affinity-based electrochemical sensors, nonenzymatic electrochemical sensors coupled with antibiofouling strategies would be a promising commercial tool used in clinical analysis for small and electroactive biological molecules which are related to human health, such as uric acid (UA), ascorbic acid (AA), dopamine (DA), guanine (G) and adenine (A). In addition, voltammetry demonstrates high selectivity and thus could be used for multi-analyte measurements. Many chemical and physical strategies have been reported to alleviate the biofouling issues, including nanoengineered surfaces (i.e., nanoporous metals and nanocarbons), antifouling layers (PEG, zwitterionic polymers, and biopolymers), nanoporous membranes, and hydrogels [4]. For example, a nanoporous gold (NPG) electrode derived from the dealloying of a silver–gold alloy was recently employed as an antibiofouling electrode for the simultaneous determination of AA and UA [5]. However, the construction of these antibioflouling electrodes is often cumbersome and complex. It would be interesting if an electrode material possessed both excellent antibiofouling performance and high electrocatalytic activity toward these molecules.

Graphene oxide (GO) is a complex carbon-based macromolecule that has a two-dimensional (2D) structure and possesses abundant oxygen-containing functional species and hydrogen atoms on its basal plane and edges. Many models have been demonstrated to elucidate the structure of GO with dazzling confusion. Recently, Siaj et al. [6] overviewed all the aspects of GO structures and presented a precise structure clarifying the chemical nature of GO based on earlier and more modern models and new discoveries by other top researchers. The structural model comprises the presence of double bond (C=C), aromatic entity, epoxyl (C-O-C), hydroxy (C-OH), carboxylic acid (O=C-OH or salt), ketone (C=O), organic carbonate, phenol, quinone, lactol, ester carbonyl, carbon vacancies, sulfate ester, carbon radicals, implicit carbon-hydrogen bond (C-H), and allylic alcohol [6]. GO exhibits excellent water dispersity and hydrophilicity due to the ionizable edge O=C-OH species. In fact, carboxylic acid distributes at the edges while phenol hydroxyl and epoxide groups mainly present on the basal plane, endowing GO an amphiphile with hydrophilic edges and a more hydrophobic basal plane [7]. The versatile surface chemistry also offers GO a powerful platform for further construction with various chemical moieties (organic, inorganic, and nanocomposites) by covalent and noncovalent interactions [8].

Unfortunately, these oxo functionalities present on GO result in poor conductivity. In efforts tailoring GO with high conductivity for applications in electrochemistry and electronics, tremendous efforts have been directed toward the elimination of the oxidized functionalities from GO by means of microorganisms, chemicals, electrochemistry, heat, UV, microwave irradiation, ion bombardments or multiphase methods [9,10,11]. Although the removal of oxygen-containing functional species increases the conductivity of the 2D material, it was recently found that the residual oxo functionalities play an essential role in promoting the electrocatalytic reaction. For example, mildly reduced GO with high concentrations of epoxy or ring ether groups located either on their basal planes or at plane edges exhibit excellent activity, high selectivity and stability for electrochemical H_2_O_2_ production from oxygen [12,13]. Furthermore, carbon defects related to quinone/catechol groups or carboxylic acid edge sites play a more pivotal role in boosting 2 e^−^ ORR peroxide formation activity than other oxygen-containing functional species under alkaline conditions for nitrogen-doped reduced GO [14]. The residual oxygen-containing functional species present on graphene also enhanced the electrocatalytic oxidation of glutathione [15], dihydroxybenzene isomers, L-methionine [16], and ascorbic acid [17], as well as the reduction of polyphenol [18]. 

It has been proved that electrochemical reduction is an ideal method to tune both conductivity and electrocatalytic activity of GO under different degrees of reduction [19]. Here, electrochemically tuned GO was explored for the electrochemical sensing of multi-analytes, and the effect of oxygen-containing functional species on both electrocatalysis and antibiofouling ability was also investigated in detail. The results showed that the residual oxo functionalities were beneficial for the selectivity and electro-oxidation of UA, DA, AA, G and A. In addition, the oxo functionalities were responsible for the binding of proteins in serum, which forms a coating layer. The binding layer facilitated the mass transport and did not hinder the electron transfer of electroactive probe Fe(CN)_6_^4−^/Fe(CN)_6_^3−^. 

## 2. Experimental Section

### 2.1. Chemicals and Solutions

Bovine Serum Albumin (BSA), UA, DA, AA, G, A, and Polydimethylsiloxanes (PDMS) were purchased from Sigma-Aldrich (St. Louis, MO, USA). GO was purchased from Nanjing XFNano Materials Tech Co., Ltd (Nanjing, Jiangsu, China). The screen-printed electrodes (DRP 110) were purchased from DROPSENS (Oviedo, Spain). All other chemicals were of analytical reagent grade and used without further purification. The aqueous solutions were prepared with doubly distilled water. An amount of 0.1 M pH = 7.2 PBS solution was used as the background electrolyte for measurements. The oxo functionalities present on GO were tuned by electrochemical treatment for 500 s at different potentials from −0.6 to −1.2 V using an I-T curve in acetic buffer (pH = 4). The electrochemical tuned GO electrode was signed as GO_potential_. For example, GO_−0.75V_ refers to the GO electrode that was electrochemically treated at −0.75 V. Unless otherwise specified, GO_−0.75V_ was used in the text.

### 2.2. Apparatus

The X-ray photoelectron spectroscopy (XPS) was recorded on a Thermo ESCALAB 250 Xi spectrometer fitted with a monochromatic Al Kα X-ray source. The morphologies were characterized by scanning electron microscopy (SEM) (Hitachi SU8010). All the electrochemical experiments were carried out with a CHI 750E electrochemical workstation with a conventional three-electrode system consisting of a modified GCE working electrode or screen-printed electrode, platinum coil auxiliary electrode, and Ag/AgCl (saturated KCl) reference electrode. Differential pulse voltammetry technology was used for electrochemical detection. Electrochemical impedance spectra (EIS) were carried out in 0.1 M KCl containing 5 mM Fe(CN)_6_^3−/4−^ in the frequency range of 1 MHz to 0.1 Hz at 0.24 V.

The screen-printed electrode modified with GO_−0.75V_ was integrated with a PDMS-based micro electrochemical cell for the analysis of five biological molecules and serum (Figure 1 and Figure 2). In order to prevent the leaking of the solution, two powerful magnets were used to compress the cell together with the electrode. The micro PDMS cell was fabricated according to Figure 2 in four steps, including mold construction, casting with PDMS mixture, baking, and mold removal. Two well-cut plastic tubes were concentrically fixed. The outer tube was sealed to a flat glass with instant adhesive. The inner tube was fixed on a screen-printed electrode template with adhesive, which was inserted through the sidewall of the outer tube and adhered to. The proper amount of degassed mixture of PDMS monomer and curing agent (in a 10:1 ratio) was poured into the ring part, followed by baking for 40 min at 100 °C. Finally, PDMS was peeled from the mold, and the cell body was obtained.

### 2.3. Electrode Preparation and Modification

Prior to modification, the GCE (3 mm diameter, 0.07 cm^2^) was successively polished with 1, 0.3, and 0.05 μm alumina paste to a mirror finish and then rinsed with water followed by an ultrasonic treatment in water and ethanol, respectively. The GO-modified GCE (GO/GCE), or screen-printed electrode, was prepared by dropping 5 μL or 7 μL GO suspension (1 mg/mL) on a cleaned GCE and left to dry at room temperature. 

### 2.4. Interaction between Electrode and Proteins

In order to study the interaction between the electrode and BSA, GO/GCE or GO_−0.75V_/GCE was immersed in 10 mg/mL BSA solution for 30 min, and the dipped electrode was signed with GO-BSA or GO_−0.75V_-BSA. In addition, GO/GCE or GO_potential_ was immersed in 100% serum for 30 min (GO-serum/GCE, GO_potential_-serum/GCE) to investigate the antibiofouling property. The interaction between BSA and UA was tested by immersing the electrode in 10 mg/mL BSA solution for 30 min and then rinsing with water, followed by dipping in 200 μM UA for 30 min (GO_−0.75V_-BSA-UA/GCE). 

### 2.5. Serum Sample Analysis for UA

An original human serum sample was diluted with PBS at a 1:40 *v*/*v* ratio for measurements without any other pretreatment. 

## 3. Results and Discussion

### 3.1. XPS Characterization of the Oxo Functionalities Present on GO_potential_

Over-reduction of GO will lose oxo functionalities, which have been proved to be beneficial for electrocatalysis. Good balance of conductivity and oxo functionalities of GO would play an important part in nonenzymatic electrochemical sensing. XPS was used for the analysis of oxo functionalities present on GO which was electrochemically treated at different potentials from −0.6 to −1 V. Figure 1 shows the recorded C1s spectra of XPS for electrochemically treated GO/GC. Different functional species, including C=C, C-C, C-OH, C-O-C, C=O, and O-C=O, were observed [19]. In order to quantify the change in different components, the content of different functionalities present on GO_potential_ versus potential is shown in Figure 2. It could be seen in Figure 2A, with the negative shift of the potential, the content of both C=O and C-O-C showed a downward trend, and C=O species disappeared when the potential was much lower than −0.8 V. The content of both C-OH and O-C=O species increased until −0.85 V, which might be attributed to the reduction of C=O, and then decreased when the potential was much lower than −0.85 V, indicating the reduction of the C-OH and O-C=O species [19].

### 3.2. Electrochemical Sensing of Multiple-Analyte

Electrochemically treated GO at different potentials was explored for the electrocatalytic oxidation of UA, DA and AA. These three molecules are electrooxidizable constituents that are commonly present in physiological fluids. It is hard to achieve efficient discrimination from each other by voltammetry due to their overlapped oxidation potential. Figure 3 shows the DPV of the mixture of 200 µM AA, 20 µM DA, and 20 µM UA in 0.1 M pH 7.2 PBS at different electrodes. Only one oxidation peak was observed for bare GCE, and two oxidation waves appeared for GO/GCE. However, when GO was electrochemically treated even at −0.6 V, three oxidation peaks with high selectivity occurred, indicating improved electrocatalytic activity. With the negative shift of the potential, the oxidation current increased until −0.75 V. Nevertheless, electrochemical treatment at more negative potential than −0.75 V would lead to decreased selectivity. This might be caused by the loss of oxo functionalities.

Although the over-electro-reduction of GO can improve the conductivity, useful oxo functionalities will be lost, which may be beneficial for electrocatalysis and selectivity. Conductivity, electro-active sites and binding interaction play an important part in electrochemical sensing. At pH = 7.2, UA exists in the anionic form (pKa = 5.4), which is more hydrophobic than AA and DA due to its very limited solubility in water [20]. AA exists as an anion (pKa = 4.10) while DA is in the cationic form (pKb = 8.87) [21]. Among these oxo functionalities present on GO, ionizable O=C-OH species distributes at the edges, while phenol hydroxyl and epoxide groups mainly present on the basal plane, leading to an edge-to-center distribution of hydrophilic and hydrophobic domains. Pristine GO/GCE exhibited much higher selectivity than bare GCE, suggesting that the oxo functionalities might interact with the analyte and consequently lead to enhanced selectivity, even though GCE has much higher conductivity than pristine GO. In summary, we speculated that the enhanced selectivity was attributed to the hydrophobic interaction between the electrode and UA and the electrostatic force between O=C-OH species and DA. In addition, conductivity and electrocatalytic activity from C=O species might also play an important role in the electrochemical behavior of UA, DA and AA.

Under optimum conditions, GO_−0.75V_/GCE was employed for the electrochemical detection of UA in the presence of AA and DA in 0.1 M pH 7.2 PBS (Appendix A). Linear calibration curves are obtained over the range of 0.2–20 µM with the calculated detection limit of 0.12 µM. Similarly, Appendix A demonstrates the electrochemical detection of DA in the presence of AA and UA and gives a linear dependence of the peak current for DA with concentrations ranging from 1 to 100 µM with the calculated detection limit of 0.5 µM.

Electrochemically treated GO at different potentials was also demonstrated for the electrocatalytic oxidation of G and A (Appendix A). GCE showed two anodic waves located at 0.63 and 0.97 V, respectively. For pristine GO-modified electrodes, the two peaks shift to 0.61 and 0.93 V, respectively. When the potential shifted negatively, the oxidation potential of G and A decreased gradually until −0.75 V. The oxidation peak of G and A shifted to 0.55 and 0.85 V at GO_−0.75V_/GCE, respectively. The decreased overpotential of G and A indicated the enhanced electrocatalytic activity, which is the result of the balance of conductivity and some oxo functionalities as seen in C=O species. Upon the potential beyond −0.75 V, the background noise increased. Therefore, GO_−0.75V_/GCE was selected for the electrochemical measurements. Appendix A present the electrochemical detection of G and A in 0.1 M pH 7.2 PBS, respectively. The calibration plot shows that the peak current of A (or G) exhibits a linear relationship (R^2^ = 0.998 or 0.999) over a concentration range from 2 µM to 100 µM (0.5 to 100 µM) with a detection limit of 0.3 µM (0.2 µM). Table 1 lists the electrochemical sensing performances for five biological molecules based on GO_−0.75V_ and other electrocatalysts. The results showed that this method exhibited a comparable linear range and detection limit with other electrocatalysts.

Next, this method was demonstrated to selectively discriminate UA, DA, AA, G, and A by using a GO_−0.75V_-modified screen-printed electrode couple with a homemade PDMS micro electrochemical cell (Figure 4). The results showed that the electrode exhibited high-resolution discrimination for five biological molecules. 

### 3.3. Antibiofouling Property

UA derives from the final metabolism of purine, which is a constituent of nucleic acids, and is finally excreted into the urine by the kidney [49]. Evaluated serum UA level (also called hyperuricemia) could lead to gout and nephrolithiasis, which has recently been found to be related to hypertension, coronary heart disease, heart failure, atrial fibrillation, insulin resistance, and nonalcoholic fatty liver disease [50]. In order to achieve the practical detection of UA in serum, the electrode should possess not only high selectivity, but also excellent antibiofouling ability. First, an Fe(CN)_6_^3−/4−^ redox couple was utilized to investigate the binding of electrodes with BSA and antibiofouling property (Figure 5). The electrodes before and after immersing in BSA exhibited distinct differences in Fe(CN)_6_^3−/4−^ solution. It is amazing to find that the current of GO-BSA/GCE is about 25 times higher than that of GO/GCE, suggesting that the BSA coating on the electrode surface does not hinder the electron transfer and mass transport. However, the current of GO_−0.75V_-BSA/GCE is only about 1.8 times higher than that of GO_−0.75V_/GCE. The findings indicate that the oxo functionalities are beneficial for the binding of BSA, which may be caused by hydrophilic interaction and hydrogen-bonding interaction. The isoionic point of BSA is about 5, which suggests that BSA exhibits a negative charge at pH 7.2 because of the loss of protons from the imidazole groups of amino acids in BSA [51]. It was considered that the binding of BSA improves the electroactive area of the electrode, and the Fe(CN)_6_^3−/4−^ redox couple can transport freely. As a result, the redox current increased dramatically when BSA was attached to the electrode. EIS was carried out to investigate the electrochemical activity of the electrodes, as shown in Appendix A. For GO/GCE and GO_−0.75V_/GCE, the equivalent circuit fitted from the EIS includes a series combination of R1, R2, Ws, CPE1 and CPE2. The electrode surface can be considered as a graphene surface that is covered by the oxygen-containing functional species [52]. In this case, R2 and CPE1 correspond to the graphene surface without oxo functionalities, and CPE2 and Ws correspond to the oxidized surface. R1 and R2 represent the electrolyte resistance and the charge transfer resistance, respectively. Ws is a finite-length Warburg element that represents the diffusion through an oxidized graphene surface. However, additional (R3, CPE3) was observed for GO-BSA/GCE and GO_−0.75V_-BSA/GCE, which is linked with the attached BSA layer. The fitted parameters for different electrodes are listed in Appendix A. There is no significant change in the charge transfer resistance (R2) before and after the modification of BSA. However, diffusion resistance (W1-R) decreased when BSA was attached to the surface, indicating that the loading of BSA improved the diffusion of Fe(CN)_6_^3−/4−^. 

Similar results were found at 100% serum modified electrode due to the binding of serum proteins (Appendix A). When the potential is more negative than −0.85 V, the current of GO_potential_-serum/GCE is much lower than that of GO_potential_ /GCE. The surface morphology and composition of the electrodes and BSA-modified electrodes were also investigated by XPS (Figure 6) and SEM (Appendix A). The carbon (1 s) spectra for GO-BSA and GO_−0.75V_-BSA can be well fitted to three peaks, suggesting three components of carbon. The peak at 284.7 eV was the bulk C peak ascribed to the alkyl chain. The second peak at approximately 286.4 eV was attributed to the C-S bond, indicating that BSA might be attached to the electrode by thiol groups or disulfide (S-S) presented on BSA. The third peak located at approximately 287.9 eV indicated amide bonds or carboxyl groups. A more clear wrinkled structure was observed for GO_−0.75V_ than GO_−0.75V_-BSA/GCE, which also indicated the attachment of BSA film (Appendix A). The antibiofouling property of the electrodes was also investigated by testing the DPV of the mixture of AA, DA, and UA at the GO_−0.75V_/GCE, GO_−0.75V_-BSA/GCE and GO_−0.75V_-serum/GCE (Figure 7). It was found that the attached BSA did not interfere with the detection of AA, DA, and UA. However, GO_−0.75V_-serum/GCE exhibited a much lower current than that of GO_−0.75V_/GCE, which might be caused by other species that existed in the serum. Appendix A shows the DPV of GO_−0.75V_/GCE, GO_−0.75V_-BSA/GCE, GO_−0.75V_-serum/GCE and GO_−0.75V_-BSA-UA/GCE in background electrolyte (0.1 M pH 7.2 PBS). An obvious oxidation peak of UA was observed, indicating that UA could be attached to BSA [53].

### 3.4. Analysis of UA in Serum

The GO-based electrochemical sensor was tested to demonstrate the capacity for the detection of UA in practical human serum samples through a standard addition method (Appendix A). The serum sample was diluted with PBS at a 1:40 *v*/*v* ratio for direct measurement without any other pretreatment. The detected content of UA in real serum was 160 µM, which is much lower than that (220 µM) obtained by the commercial clinical method. 

## 4. Conclusions

A micro electrochemical sensor was developed for multi-analyte detection based on the electrochemically tuned GO electrode. The key problem of biofouling was investigated and discussed in detail. The results showed that the oxo functionalities present on GO played an important role in electrocatalysis, selectivity, and antibiofouling performance. In this work, we presented a low-cost and simple electrode that possessed both excellent antibiofouling performance and high electrocatalytic activity toward the multi-analyte. Electroactive probe investigation indicated that the attachment of BSA did not hinder the electron transfer but enhanced the diffusion of probes. This electrode was demonstrated to detect the real content of UA in a serum sample. This simple method will provide a promising potential in clinical application. 

## Data Availability

The authors confirm that the data supporting the findings of this study are available within the article [and/or] its Appendix A.

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
