# Peer review of "A Micro Electrochemical Sensor for Multi-Analyte Detection Based on Oxygenated Graphene Modified Screen-Printed Electrode"

_nanomaterials, 2022, doi:10.3390/nano12040711_

Round 1

Reviewer 1 Report

In this study, multi-analytes were detected using oxo functionalities GO electrodes. The authors explain that oxo functionalities GO electrodes are effective in reducing biofouling. In addition, novel properties of the BSA modified electrode were introduced. AA, DA, UA, G, and A were detected using this, and detection in real serum was also performed. However, the study results raise the following questions. Also, there are too many small mistakes throughout the manuscript. In order to be published as an article on Nanomaterials, it is necessary to answer the following questions.

  1. As shown in Fig. 10 and 11, the increase in the electrochemical activity of the electrode when BAS (or serum) is modified is the opposite result of generally known. In typical, if the protein is modified on the electrode surface, the electrochemical activity of the electrode decreases. (Microchim Acta 184 (2017) 1353–1360, Talanta 79 (2009) 159–164) It is necessary to add the impedance data and explain in detail why these results were obtained.
  2. In Fig. 16, how was 'the detected content of UA in real serum' calculated to be 160 μM? Also, what is the commercial clinical method?
  3. In Table 1., add the detection time and kind of sample such as just buffer, serum, saliva, and etc. to make a more practical and accurate comparison.
  4. There should be a space between numbers and units. Check and revise the entire manuscript.
  5. About 6 figures in the manuscript are suitable. Move the remaining figures to Supplementary Material.
  6. Make sure all references are in the same form. Follow the journal's template. In particular, modify the form of the abbreviation of the journal to be the same.
  7. Explain Fig. more clearly in the caption. Describe the detailed conditions of the experiment obtained in Fig. For example, information of buffer, chemical concentration, reaction time, and important parameters of DPVs and CVs are required.
  8. In the caption, change Scheme 1 and Scheme 2 to bold style.

Reviewer 2 Report

 There are already many papers measuring similar redox potentials for UA, AA, and DA, but they are not cited. Although the author's method insisted that GO sensor has a high resolution of identification, the uncited papers already have excellent separation and quantification examples.

 From the results of the XPS peak separation of GO alone, the estimation of the adsorption of the object to be measured has been stated for a long time, and there is no citation in the literature, and there is almost no evidence.

The source of commercially available screen-printed electrodes is not clear.

Measurements in serum did not clearly assess how the coexistence of BSA affects the actual measurement method. So what percentage of the measurement error is caused by the coexistence of BSA? Also, there are no citations from other papers on this subject and they cannot be compared.

Author Response

We greatly appreciate Reviewer 1 for the constructive suggestion. We revised the Introduction and Conclusion sections.

Round 2

Reviewer 1 Report

The manuscript has been significantly improved. The figures were well organized and the research content was well explained.

Reviewer 2 Report

This paper is not intended for journals of analytical chemistry, so pursuing the practicality of this method may be out of scope. However, I thought there was still a problem with the actual sensor 3.4 application. I would like you to investigate the cause and make further improvements.

Reviewer 3 Report

Dear Editor,

the authors revised the manuscript. I think it has improved from the original, although not as much as I think the writers could have. In particular, I think the discussion is still weak but, as a reviewer, I can suggest changes and not impose them. I have no particular reasons to insist but there remains a bit of disappointment for a paper that had the opportunity to stand out from many others of the same type. It can also be published as is.

Best regards